# SeAttE: An Embedding Model Based on Separating Attribute Space for Knowledge Graph Completion

Zongwei Liang *,† , Junan Yang † , Hui Liu, Keju Huang, Lingzhi Qu, Lin Cui and Xiang Li





College of Electronic Engineering, National University of Defense Technology, Hefei 230037, China; yangjunan@ustc.edu.cn (J.Y.); christ592604@163.com (H.L.); huangkeju@163.com (K.H.); qulingzhi@nudt.edu.cn (L.Q.); cuilin17@nudt.edu.cn (L.C.); lix20@nudt.edu.cn (X.L.)
\* Correspondence: zwliang17@nudt.edu.cn
† These authors contributed equally to this work.

**Abstract:** Knowledge graphs are structured representations of real world facts. However, they typically contain only a small subset of all possible facts. Link prediction is the task of inferring missing facts based on existing ones. Knowledge graph embedding, representing entities and relations in the knowledge graphs with high-dimensional vectors, has made significant progress in link prediction. The tensor decomposition models are an embedding family with good performance in link prediction. The previous tensor decomposition models do not consider the problem of attribute separation. These models mainly explore particular regularization to improve performance. No matter how sophisticated the design of tensor decomposition models is, the performance is theoretically under the basic tensor decomposition model. Moreover, the unnoticed task of attribute separation in the traditional models is just handed over to the training. However, the amount of parameters for this task is tremendous, and the model is prone to overfitting. We investigate the design approaching the theoretical performance of tensor decomposition models in this paper. The observation that measuring the rationality of specific triples means comparing the matching degree of the specific attributes associated with the relations is well-known. Therefore, the comparison of actual triples needs first to separate specific attribute dimensions, which is ignored by existing models. Inspired by this observation, we design a novel tensor ecomposition model based on Separating Attribute space for knowledge graph completion (SeAttE). The major novelty of this paper is that SeAttE is the first model among the tensor decomposition family to consider the attribute space separation task. Furthermore, SeAttE transforms the learning of too many parameters for the attribute space separation task into the structure's design. This operation allows the model to focus on learning the semantic equivalence between relations, causing the performance to approach the theoretical limit. We also prove that RESCAL, DisMult and ComplEx are special cases of SeAttE in this paper. Furthermore, we classify existing tensor decomposition models for subsequent researchers. Experiments on the benchmark datasets show that SeAttE has achieved state-of-the-art among tensor decomposition models.

**Keywords:** NLP; knowledge graphs; knowledge representation; link prediction; attribute space; separation

## 1. Introduction

Knowledge Graphs (KGs) are collections of large-scale triples, such as Freebase [1], YAGO [2] and DBpedia [3]. KGs play a crucial role in applications such as question answering services, search engines, and smart medical care. Although there are billions of triples in KGs, they are still incomplete. These incomplete knowledge bases will bring limitations to practical applications [4]. For example, over 70% of people included in Freebase have no known place of birth,and 99% have no known ethnicity, which will significantly limit our search and answering [5]. Therefore, knowledge graph completion,

known as link prediction, which automatically predicts missing links between entities based on given links, has recently attracted growing attention.

Inspired by word embedding [6], researchers recently tried to solve the task of link prediction through knowledge graph embedding. Knowledge graph embedding models map entities and relations into low-dimensional vectors (or matrices, tensors), measure the rationality of triples through specific score functions between entities and relations, and rank the triples with scores. TransE [1] first proposes to utilize relation vectors as the geometric distance between entities. Then many variants emerge.

The tensor decomposition models [7–13] are a family of which the inference performance is relatively good among these variants. RESCAL [7] is the basic tensor decomposition model, which is the first tensor decomposition model. Since RESCAL [7] represents the relations as a matrix, the large number of parameters makes it difficult for the model to learn effectively. So DisMult [8] directly diagonalizes the matrix, which takes the relations as vectors. This operation significantly reduces the number of parameters. There are a large number of complex relation types in the knowledge graphs. However, DisMult is an over-simplified model, which cannot describe complex relations. Then subsequent variants are invented to describe more types of relations, such as asymmetric and hierarchical relations, which are equivalent to designing unique structures for description of specific types of relations. For example, ComplEx [9], similarly to DistMult [8], forces each relation embedding to be a diagonal matrix but extends such formulation in the complex space. Analogy [14] aims at modeling analogical reasoning, which is crucial for any knowledge induction. It employs the general bilinear scoring function but adds two main constraints inspired by analogical structures. TuckER [10] relies on the Tucker decomposition [15], which factorizes a tensor into a set of vectors and a smaller shared core. SimplE [11] forces relation embeddings to be diagonal matrices, similarly to DistMult [8], but extends it by associating two separate embeddings with each entity and associating two separate diagonal matrices with each relation. These models mainly explore particular regularization to improve performance. No matter how sophisticated the design of such tensor decomposition models is, they find it difficult to surpass the basic tensor decomposition model theoretically. In addition, the previous tensor decomposition models do not consider the problem of attribute separation. The unnoticed task of attribute separation in the traditional models is just handed over to the training. However, the amount of parameters for this task is tremendous, and the model is prone to overfitting.

Considering that none of the variant models under the current research route can exceed the theoretical tensor decomposition model, we focus on making the tensor decomposition model approach the theoretical performance in this paper. The tensor decomposition models cannot achieve theoretical performance because too many parameters limit the dimensional expansion. Inspired by attribute selection in practical comparisons of triples, we propose a tensor decomposition model based on attribute subspace segmentation in this paper.

In practice, entities are collections of attributes, and different entities can contain various semantic attributes. Comparing triples with different relations should only select specific attributes for comparison. Figure 1 shows the comparison of boxes with the same shape and different colors. When comparing different attributes such as colors or shapes, we should first separate the colors or shapes of the entities that need to be compared and then compare the associations of the corresponding colors or shapes of the entities. Inspired by this fact, we should first separate the properties that need to be compared. Measuring the plausibility of a given triple means comparing the matching degree of the attributes associated with the predicate between the entities. However, the traditional tensor decomposition model ignores the first operation (attribute separation). Therefore, we propose a novel model—a tensor decomposition model based on separating attribute space for knowledge graph completion (SeAttE) in this paper. SeAttE transfers the large-parameter learning for the attribute space separation task in traditional tensor decomposition models to the model structure design. This operation effectively reduces the

number of parameters, allowing the model to focus on learning the semantic equivalence between relations and better performance.

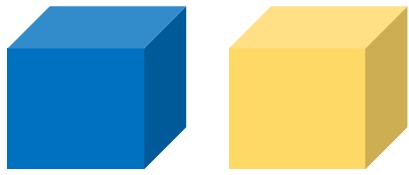

**Shape**: Cube   $=$   Cube

**Colour**: Blue   $\neq$   Yellow

**Figure 1.** Comparison of boxes with the same shape and different colors.

The actual size of the attribute subspace is related to the complexity of the relations. Predefined designs cannot accurately model the relations. In order to facilitate the realization of the model, we propose the initialization design of the uniform attribute subspace. Specifically, SeAttE limits the size of each attribute subspace by setting the maximum attribute subspace dimension. In this paper, the large amount of parameters that need to be learned for the attribute space separation task is transformed into the design of the model structure. This design dramatically reduces the need to learn parameters so that the tensor decomposition model can be extended to higher dimensions, significantly improving performance.

Overall, inspired by the fact that inference should first perform attribute space filtering, we propose SeAttE—a tensor factorization model based on separating attribute space for knowledge graph completion in this paper. Our main contributions are as follows.

- SeAttE is the first model among the tensor decomposition family to consider the attribute space separation task. SeAttE transforms the learning of too many parameters for the attribute space separation task into the structure's design. This operation allows the model to focus on learning the semantic equivalence between relations, causing the performance to approach the theoretical limit. Experiments on the benchmark datasets show that SeAttE achieves state-of-the-art among the tensor factorization models.
- We prove that RESCAL, DisMult, and ComplEx are all special cases of SeAttE in this paper;
- We classify the tensor factorization models from a new perspective for their better understanding by subsequent researchers.

The rest of this paper is organized as follows: Section 2 presents a brief overview of related work. We provide the problem formulation, including definitions, preliminaries and research questions in Section 3. We analyze the design of SeAttE and prove the relation to previous tensor factorization models in Section 4. The experiments are conducted and discussed with the existing KG embedding models in Section 5. Finally, we summarize our findings along with the future directions in Section 6.

## 2. Related Work

In this section, we describe related works and the critical differences between them. We divide knowledge graph embedding models into three leading families [16–19], including Tensor Decomposition Models, Geometric Models, and Deep Learning Models.

**Tensor Decomposition Models.** These models implicitly consider triples as tensor decomposition. DistMult [8] constrains all relation embeddings to be diagonal matrices, which reduces the space of parameters to access a more accessible model to train. RESCAL [7] represents each relationship with a total rank matrix. ComplEx [9] extends the KG embeddings to the complex space to better model asymmetric and inverse relations. Analogy [14]

employs the general bilinear scoring function but adds two main constraints inspired by analogical structures. Based on the Tucker decomposition, TuckER [10] factorizes a tensor into a set of vectors and a smaller shared core matrix. SimplE [11] is a simple enhancement of CP to allow the two embeddings of each entity to be learned dependently. HolE [13] is a multiplicative model that is isomorphic to ComplEx [9]. Inspired by the recent success of automated machine learning (AutoML), AutoSF [12] proposes to automatically design scoring functions for distinct KGs by the AutoML techniques. QuatDE [20] captures the variety of relational patterns and separates different semantic information of the entity, using transition vectors to adjust the point position of the entity embedding vectors in the quaternion space via Hamilton product, enhancing the feature interaction capability between elements of the triplet. DensE [21] develops a novel knowledge graph embedding method to provide an improved modeling scheme for the complex composition patterns of relations.

**Geometric Models.** Geometric Models interpret relations as geometric transformations in the latent space. TransE [1] is the first translation-based method, which treats relations as translation operations from the head entities to the tail entities. Along with TransE [1], multiple variants, including TransH [22], TransR [23] and TransD [24], are proposed to improve the embedding performance of KGs. Recently, RotatE [25] defines each relation as a rotation from head entities to tail entities. Inspired by the fact that concentric circles in the polar coordinate system can naturally reflect the hierarchy, HAKE [26] maps entities into the polar coordinate system. HAKE [26] can effectively model the semantic hierarchies in knowledge graphs. OTE [27] proposes a distance-based knowledge graph embedding. First, OTE extends the modeling of RotatE from 2D complex domain to high dimensional space with orthogonal relation transforms. Second, graph context is proposed to integrate graph structure information into the distance scoring function to measure the plausibility of the triples during training and inference.

**Deep Learning Models.** Deep Learning Models use deep neural networks to perform knowledge graph completion. ConvE [28] and ConvKB [29] employ convolutional neural networks to define score functions. CapsE [30] embeds entities and relations into one-dimensional vectors under the basic assumption that different embeddings encode homologous aspects in the same positions. CompGCN [31] utilizes graph convolutional networks to update the knowledge graph embedding. Neural Tensor Network (NTN) combines E-MLP with several bilinear parts. Nathani [32] proposes a novel attention-based feature embedding that captures both entity and relation features in any given entity's neighborhood. RLH [33] is inspired by the hierarchical structure through which a human being handles cognitionally ambiguous cases. The whole reasoning process is decomposed into a hierarchy of two-level Reinforcement Learning policies for encoding historical information and learning structured action space. R2D2 [34] is a novel method for automatic reasoning on knowledge graphs based on debate dynamics. R2D2 is to frame the task of triple classification as a debate game between two reinforcement learning agents which extract arguments-paths in the knowledge graph—with the goal to promote the fact being true (thesis) or the fact being false (antithesis), respectively. RNNLogic [35] is a probabilistic model. RNNLogic treats logic rules as a latent variable, and simultaneously trains a rule generator as well as a reasoning predictor with logic rules. MADLINK [36] introduces an attentive encoder–decoder-based link prediction approach considering both structural information of the KG and the textual entity descriptions.

There are also other models, such as DURA [37], which are proposed to solve overfitting. RuleGuider [38] leverages high-quality rules generated by symbolic-based methods to provide reward supervision for walk-based agents. SFBR [39] provides a relation-based semantic filter to extract the attributes that need to be compared and suppress the irrelevant attributes of entities. Together, most of the above studies intend to find a more robust representing approach. Measuring the effectiveness of certain triples is to compare the matching degree of specific attributes based on relations. Only a few models, such as TransH [22], TransR [23], and TransD [24], consider that entities in different triples should

have different representation. However, these variants require many resources occupations and are limited to particular models.

Although there is much research on this task, this paper mainly focuses on the models based on tensor decomposition. The previous tensor decomposition models mainly achieved better performance through unique regularization, but these models still could not reach the theoretical upper limit of the tensor decomposition model. No matter how sophisticated the design of tensor decomposition models is, the performance is theoretically under the basic tensor decomposition model. Moreover, the previous tensor decomposition model did not consider the problem of attribute separation. The unnoticed task of attribute separation in the traditional models was just handed over to the training. However, the amount of parameters for this task is tremendous, and the model is prone to overfitting. Inspired by the actual semantic comparison, this paper proposes an attribute subspace structure design—SeAttE, which reaches the theoretical upper limit of the tensor decomposition model. We will describe the relationship between SeAttE and other models based on tensor decomposition in detail in Section 4.3.

## 3. Background

In this section, we introduce KG embedding, KG completion tasks and the notations used throughout this paper. Next, we briefly introduce several models involved in this paper.

### 3.1. KG Completion and Notations

KGs are collections of factual triples $K = \{(h, r, t), h, t \in \mathcal{E}, r \in \mathcal{R}\}$, where $(h, r, t)$ represents a triple in the knowledge graph, $h, t, r$ are head, tail entities and relations, respectively. We associates the entities $h, t$ and relations $r$ with vectors $\mathbf{h}, \mathbf{t}, \mathbf{r} \in \mathbf{R}^d$ in knowledge graph embedding. Then we design an appropriate scoring function $d_r(\mathbf{h}, \mathbf{t})$: $\mathcal{E} \times \mathcal{R} \times \mathcal{E} \rightarrow \mathbf{R}$, to map the embedding of the triple to a certain score. For a particular question $(h, r, ?)$, the task of KG completion is ranking all possible answers and obtain the preference of prediction.

We use $\mathbf{W_r} \in \mathbf{R}^{d \times d}$ and $\mathbf{r} \in \mathbf{R}^d$ to distinguish matrix representation and vector representation of the relations, respectively. $T$, $\langle \cdot \rangle$ and $\circ$ denote the operation of transpose, the generalized dot product and the Hadamard product, respectively. Especially, we utilize $r_{SeAttE}$ to represent the matrix of relation in SeAttE. Let $\|\|$, diag() and $\mathbf{Re}()$ denote the $L_2$ norm, matrix diagonalization and the real part of complex vectors.

### 3.2. Basic Models

**Tensor Factorization Models.** Models in this family interpret link prediction as a task of tensor decomposition, where triples are decomposed into a combination (e.g., a multi-linear product) of low-dimensional vectors for entities and relations. CP [40] represents triples with canonical decomposition. Note that the same entity has different representations at the head and tail of the triplet. The score function can be expressed as:

$$d_r(\mathbf{h}, \mathbf{t}) = \left\| \mathbf{h}^{\mathbf{T}} \mathbf{r} \mathbf{t} \right\| \tag{1}$$

where $\mathbf{h}, \mathbf{r}, \mathbf{t} \in \mathbf{R}^k$.

RESCAL [7] represents a relation as a matrix $\mathbf{W_r} \in \mathbf{R}^{d \times d}$ that describes the interactions between latent representations of entities. The score function is defined as:

$$d_r(\mathbf{h}, \mathbf{t}) = \left\| \mathbf{h}^{\mathbf{T}} \mathbf{W_r} \mathbf{t} \right\| \tag{2}$$

DistMult [8] forces all relations to be diagonal matrices, which consistently reduces the space of parameters to be learned, resulting in a much easier model to train. On the

other hand, this makes the scoring function commutative, which amounts to treating all relations as symmetric.

$$d_r(\mathbf{h}, \mathbf{t}) = \left\| \mathbf{h}^\mathbf{T} \mathbf{W_r} \mathbf{t} \right\| \tag{3}$$

where $\mathbf{W_r} = \mathbf{diag}(\mathbf{w_1}, \mathbf{w_2}, \ldots, \mathbf{w_n})$.

ComplEx [9] extends the real space to complex spaces and constrains the embeddings for relation to be a diagonal matrix. The bilinear product becomes a Hermitian product in complex spaces. The score function can be expressed as:

$$d_r(\mathbf{h}, \mathbf{t}) = \mathbf{Re}\left(\mathbf{h}^\mathbf{T} \mathrm{diag}(\mathbf{r})\mathbf{t}\right) \tag{4}$$

where $\mathbf{h}, \mathbf{r}, \mathbf{t} \in \mathbf{C}^k$.

## 4. SeAttE Model

This section introduces a novel model—an **E**mbedding model based on **Se**parating **Att**ribute space for knowledge graph completion. We first introduce the motivation and the specific design of SeAttE in Section 4.1 and the relation to previous models in Section 4.2. Finally, we classify the current tensor factorization models in Section 4.3.

### 4.1. Motivation and Design of SeAttE

We first analyze the design route of the current models and then introduce the motivation of SeAttE in the Section 4.1.1, Then we introduce the specific design of SeAttE in Section 4.1.2.

#### 4.1.1. Motivation

As shown in Figure 2, RESCAL is the basic tensor decomposition model. Since RESCAL represents the relations as a matrix, the large number of parameters makes it difficult for the model to learn effectively. So DisMult directly diagonalizes the matrix, significantly reducing the number of parameters. However, over-simplified models limit the performance. Subsequently, variants are invented for describing specific types of relations, such as asymmetric and hierarchical relations, which are equivalent to designing unique structures for describing specific types of relationships. Such models need to look for special functions to precisely fit different relations categories. Some relations can be well characterized in models, while some are not. This design from a specific relationship type is challenging to cover all relations. No matter how sophisticated the design of such models is, it is difficult to surpass the RESCAL model theoretically. Moreover, the previous tensor decomposition model did not consider the problem of attribute separation. The unnoticed task of attribute separation in the traditional models is just handed over to the training. However, the amount of parameters for this task is tremendous, and the model is prone to overfitting.

It is widely accepted that each entity contains different attributes, and the relations describe the association of entities on specific attributes. When comparing the plausibility of triples, the first step is to pick out the semantic dimension that the relation compares and filter out irrelevant dimensions. In the second step, we compare the correlation of the attributes of heads and tails under specific attributes, whether it satisfies the triples. It is essential to separate the dimensions that need to be compared from those unrelated dimensions. However, existing tensor decomposition models ignore the isolation of attribute dimensions, and these models combine these two steps for training. These models simultaneously complete the separation of attributes and the learning of semantic equivalence. This combination will result in too many parameters for learning. Therefore, we make a unique design for the relation matrix based on the subspace theory so that the different semantic spaces will not overlap. The model implements the isolation of different attributes in the structural design.

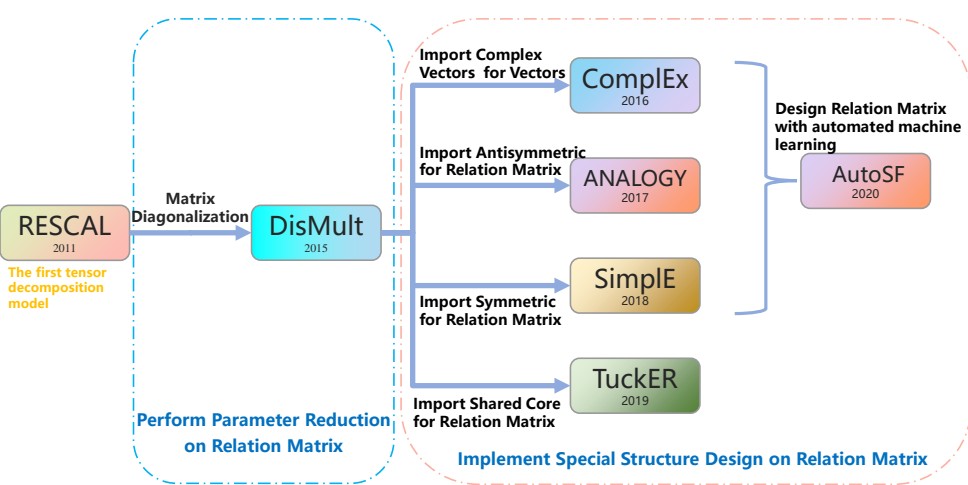

**Figure 2.** The research routes of current tensor decomposition models.

As shown in Figure 3, the left is the traditional entity vector and relation matrix; the right is the entity vector and the relation matrix with the separation of attribute spaces. We perform vector subspace separation on the relation matrix of tensor decomposition models. As shown in Equation (5), the task of attribute isolation is transferred to the model structure design. This operation allows the model to focus on learning the semantic equivalence between relations, resulting in better performance. Since the model is a new embedding model that separates attribute space for knowledge graph completion, we name the model SeAttE in this paper.

$$
\begin{aligned}
d_r(h,t) &= \|h \times r \times t\| \\
\Rightarrow d_r(h,t) &= \|h \times r_{SeAttE} \times t\|
\end{aligned}
\tag{5}
$$

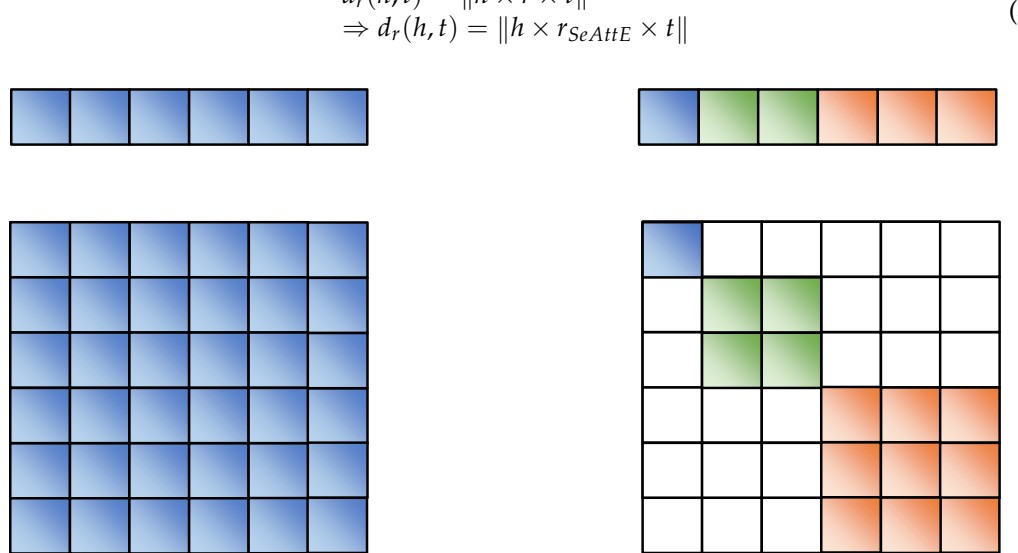

**Figure 3.** The left is the traditional entity vector and relation matrix, the right is the entity vector and the relation matrix with the separation of attribute spaces.

### 4.1.2. Design

In theory, the subspace separation should be related to the actual relations, which cannot be designed in advance. We design the structure of attribute subspace segmentation to reduce the model's workload in learning segmentation tasks of different semantic dimensions.

In order to facilitate the design and implementation of the model, SeAttE adopts the exact size of attribute subspace design. Assuming that the dimension of each entity vector is $d$ and the dimension of each attribute subspace is $k$, each entity contains $d/k$ attribute spaces.

$$r_{SeAttE} = \begin{vmatrix} W_1 & 0 & 0 & 0 \\ 0 & W_2 & 0 & 0 \\ 0 & 0 & \cdots & 0 \\ 0 & 0 & 0 & W_k \end{vmatrix} \tag{6}$$

where $r_{SeAttE} \in \mathbf{R}^d$, $W_k \in \mathbf{R}^k$ and $h = d/k$.

As shown in the left part of Figure 4, the dimension of each entity vector $d$ is eight, and the dimension of each attribute subspace $k$ is two, then the entity contains four attributes subspaces. As shown in the right part of Figure 4, when the dimension of each attribute subspace $d$ is two and the dimension of each subspace $k$ is four, the entity contains two attributes subspaces.

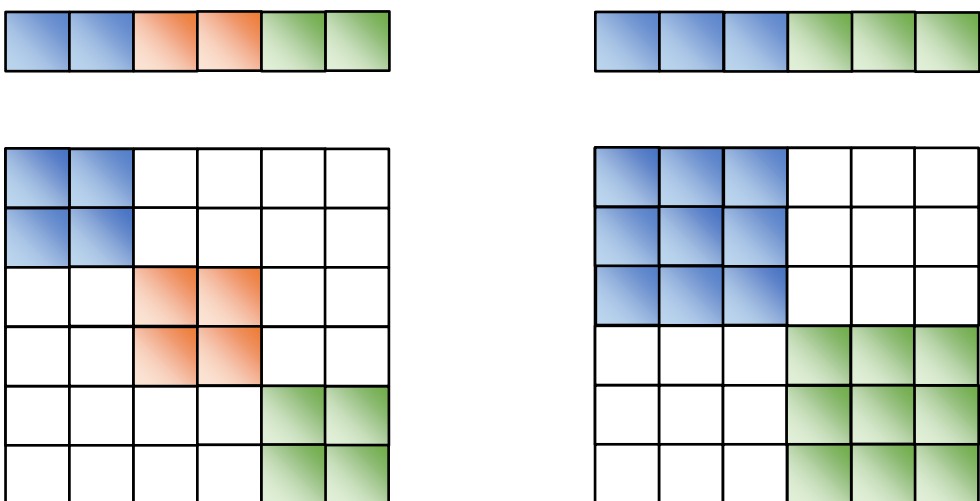

**Figure 4.** Different attribute subspace sizes under the same entity dimension. The dimension of each attribute subspace is set to 2 in the left and 3 in the right.

SeAttE realizes the division of knowledge graph attribute space by setting the max dimension of the attribute subspace. The model avoids a large number of parameter learning for attribute separations by setting the parameter of the maximum semantic space dimension.

*4.2. Relation to Previous Tensor Factorization Models*

This subsection mainly analyzes the relationship between SeAttE and traditional tensor decomposition models.

**RESCAL** is the basic tensor decomposition model. Due to the tremendous amount of parameters of this model, the dimension of the entity cannot be well expanded. When the dimension of the attribute subspace of SeAttE satisfies $k = d$, SeAttE is equivalent to RESCAL.

$$r_{SeAttE} = |W_1| \tag{7}$$

where $k = d$ and $h = 1$.

**DisMult** is the simplest tensor decomposition model, which diagonalizes all relation matrices. When the max dimension of the attribute subspace of SeAttE $k$ is set to 1, then $W_k$ is a 1-dimensional matrix, that is, a numerical value. The relationship matrix is equivalent to the diagonal. Under these circumstances, SeAttE is equivalent to DisMult.

$$r_{SeAttE} = \begin{vmatrix} W_1 & 0 & 0 & 0 \\ 0 & W_2 & 0 & 0 \\ 0 & 0 & \cdots & 0 \\ 0 & 0 & 0 & W_h \end{vmatrix} \tag{8}$$

$$= diag(W_1, W_2, \cdots, W_h)$$

where $W_k \in \mathbf{R}$.

**ComplEx** imports complex representations to characterize symmetric and antisymmetric relations.

$$
\begin{aligned}
d_r(s,o) &= \mathrm{Re}(\langle w_r, e_s, e_o \rangle) \\
&= \mathrm{Re}\left( \sum_{k=1}^{K} w_{rk} e_{sk} \bar{e}_{ok} \right) \\
&= \mathrm{Re}(w_r)\mathrm{Re}(e_s)\mathrm{Re}^T(e_o) + \mathrm{Re}(w_r)\mathrm{Im}(e_s)\mathrm{Im}^T(e_o) \\
&\quad + \mathrm{Im}(w_r)\mathrm{Re}(e_s)\mathrm{Im}^T(e_o) - \mathrm{Im}(w_r)\mathrm{Im}(e_s)\mathrm{Re}^T(e_o) \\
&= [\mathrm{Re}(e_s) || \mathrm{Im}(e_s)] W_r [\mathrm{Re}(e_o) || \mathrm{Im}(e_o)]^T \\
&= e'_s W_r (e'_o)^T
\end{aligned}
\tag{9}
$$

$$
W_r = \begin{bmatrix} diag(\mathrm{Re}(w_r)) & diag(\mathrm{Im}(w_r)) \\ diag(-\mathrm{Im}(w_r)) & diag(\mathrm{Re}(w_r)) \end{bmatrix}
\tag{10}
$$

where $e'_s, e_o' \in R^{2K}$ and $W_r \in R^{2K \times 2K}$.

From the above formula, we can find that ComplEx is equivalent to RESCAL with $d = 2k$. The model performs a particular regularization for each relation matrix, which only retains the diagonal elements of the four sub-matrices of the matrix, and the remaining elements are set to 0.

When the dimension of the attribute subspace of the SeAttE model $k$ is set to 2, the relation matrix can also be expressed as the following.

$$
\begin{aligned}
r_{SeAttE} &= \begin{vmatrix} W_1 & O & O & O \\ O & W_2 & O & O \\ O & O & \cdots & O \\ O & O & O & W_h \end{vmatrix} \\
&= \begin{vmatrix} \begin{matrix} W_{11} & W_{12} & 0 & 0 \\ W_{13} & W_{14} & 0 & 0 \\ 0 & 0 & W_{21} & W_{22} \\ 0 & 0 & W_{23} & W_{24} \end{matrix} & & O & \\ & \ddots & & \begin{matrix} 0 & 0 \\ 0 & 0 \end{matrix} \\ & O & & \begin{matrix} 0 & 0 & W_{h1} & W_{h2} \\ 0 & 0 & W_{h3} & W_{h4} \end{matrix} \end{vmatrix} \\
&= H * \begin{vmatrix} diag(W_{11}, W_{21}, \cdots, W_{h1}) & diag(W_{12}, W_{22}, \cdots, W_{h2}) \\ diag(W_{13}, W_{23}, \cdots, W_{h3}) & diag(W_{14}, W_{24}, \cdots, W_{h4}) \end{vmatrix} * G
\end{aligned}
\tag{11}
$$

where $H = h_{2\_n+1} \times h_{3\_n+2} \times \cdots \times h_{n\_2n-1}$, $h_{i\_k}$ is obtained by exchanging the $i$-th row and the $k$-th row of the identity matrix, that is, performing elementary row transformation on the matrix $W$. Where $G = g_{2\_n+1} \times g_{3\_n+2} \times \cdots \times g_{n\_2n-1}$, $g_{i\_k}$ is obtained by exchanging the $i$-th column and the $k$-th column of the identity matrix, that is, performing elementary column transformation on the matrix $W$.

When a regularization term is applied to the relation matrix of SeAttE, namely $W_{i1} = W_{i4}$ and $W_{i2} = -W_{i3}$, SeAttE is equivalent to ComplEx. In summary, when each subspace matrix of SeAttE satisfies A and B, SeAttE is equivalent to ComplEx.

### 4.3. Classification of Tensor Decomposition Models

The current tensor decomposition models are variants based on RESCAL [7]. Furthermore, the design of all models can be understood as the regularization of the relational matrix. We classify the current tensor decomposition models from a new angle so that subsequent researchers can better understand these tensor decomposition models. According to regularization, we divide the models into three families: models based on

separating attribute space, models based on symmetric regularization and models based on an orthonormal basis.

The first family realizes the semantic space separation by the subspace segmentation of relation matrix. Such models mainly include RESCAL [7], DisMult [8], and SeAttE. RESCAL [7] and DisMult [8] are special cases of SeAttE. When the max dimension of the attribute subspace satisfies $k = d$, SeAttE is equivalent to RESCAL [7], and when the max dimension of the attribute subspace satisfies $k = 1$, SeAttE is equivalent to DisMult [8].

Models based on symmetric constraints are created by imposing symmetric or antisymmetric constraints on the relational matrix. It mainly includes ANALOGY [14] and SimplE [11].

The model based on orthonormal basis representation is TuckER. This model is exceptional. It represents the relationship matrix through the linear combination of the orthonormal basis and realizes the reduction in the parameters of the relationship matrix. This model achieves link prediction by reducing the parameters of the relationship matrix through a linear combination.

Some other models combine subspace division and symmetric regularization, including ComplEx [9] and AutoSF [12].

## 5. Experiments and Discussion

This section is organized as follows. First, we introduce the experimental settings in Section 5.1. Then, we show the effectiveness of SeAttE on three benchmark datasets in Section 5.2. Finally, we visualize and analyze the embeddings generated by SeAttE in Section 5.3.

### 5.1. Experimental Settings

**Dataset.** In order to evaluate the proposed module, we consider three common knowledge graph datasets—WN18RR [41], FB15k-237 [28] and YAGO3-10 [42]. Details of these datasets are listed in Table 1.

**Table 1.** The number of entities, relations and observed triples in each split for four benchmarks.

| Dataset | #Entity | #Relation | #Training | #Validation | #Test |
|---------|---------|-----------|-----------|-------------|-------|
| WN18RR | 40,943 | 11 | 86,835 | 3034 | 3134 |
| FB15K-237 | 14,505 | 237 | 272,115 | 17,535 | 20,466 |
| YAGO3-10 | 123,182 | 37 | 1,079,040 | 5000 | 5000 |

FB15k-237 is obtained by eliminating the inverse and equal relations in FB15K, making it more difficult for simple models to do well. WN18RR is achieved by excluding inverse and equal relations in WN18. The main relation patterns are symmetry/antisymmetry and composition. YAGO3-10 is a subset of YAGO3, which is produced to alleviate the test set leakage problem.

**Evaluation Protocol and Settings.** For evaluation, we use the same ranking procedure as in the literature [43]. For each test triple, the head is removed and replaced by each of the entities of the dictionary in turn. Dissimilarities (or energies) of those corrupted triplets are first computed by the models and then sorted by ascending order; the rank of the correct entity is finally stored. This whole procedure is repeated while removing the tail instead of the head. We use evaluation metrics standard across the link prediction literature: mean reciprocal rank (MRR) and Hits@k, k = 1,3,10. Mean reciprocal rank is the average of the inverse of the mean rank assigned to the true triple over all candidate triples. Hits@k measures the percentage of times a true triple is ranked within the top k candidate triples. We evaluate the performance of link prediction in the filtered setting [1], i.e., all known true triples are removed from the candidate set except for the current test triple. In both settings, higher MRR or higher Hits@1/3/10 indicate better performance.

**Baselines and Training Protocol.** In this section, we compare the performance of SeAttE against two categories of KGC models: (1) geometric models including TransE [1], TransH [22], TransR [23], RotatE [25], TucKer [10], AutoERTR [44]and HAKE [26]; (2) models based on tensor decomposition including CP [40], SimplE [11], DisMult [8], RESCAL [7], ANALOGY [14], ComplEx [9], DURA [37], SFBR [39] and AutoSF [12]. (3) deep learning models including ConvE [28], RAN [45] ConvKB [29], CapsE [30] and Nathani [32].

Because ComplEx is a particular case of SeAttE, the parameters of our experiments are consistent with those in DURA [37]. SeAttE only introduces the parameter of the attribute subspace dimension based on DURA, which will be marked in the specific experimental results.

### 5.2. Comparison with Existing Link Prediction Models

In this section, we compare the results of SeAttE and other state-of-the-art models on three benchmark datasets.

Table 2 shows the comparison between SeAttE and geometric models. The table shows that SeAttE outperforms all the compared geometric models in MRR, Hit@1 and Hit@1. Compared with the best geometric model—HAKE, SeAttE still has significant improvements: on YAGO3-10, MRR increases by 4%; on FB15k-237, MRR increases by 2.5%.

**Table 2.** This is the comparison between SeAttE and geometric models on WN18RR, FB15K-237 and YAGO3-10. The best results of each metric for each dataset are marked in bold.

| | WN18RR | | | FB15K-237 | | | YAGO3-10 | | |
|---|---|---|---|---|---|---|---|---|---|
| | MRR | Hit@1 | Hit@10 | MRR | Hit@1 | Hit@10 | MRR | Hit@1 | Hit@10 |
| TransE | 0.223 | 0.028 | 0.510 | 0.298 | 0.217 | 0.475 | 0.501 | 0.406 | 0.674 |
| TransH | 0.224 | - | 0.504 | 0.290 | - | 0.490 | - | - | - |
| TransR | 0.235 | - | 0.510 | 0.314 | - | 0.510 | - | - | - |
| RotatE | 0.476 | 0.428 | 0.571 | 0.338 | 0.241 | 0.533 | 0.498 | 0.405 | 0.671 |
| CrossE | 0.405 | 0.381 | 0.450 | 0.298 | 0.212 | 0.471 | 0.446 | 0.331 | 0.655 |
| TorusE | 0.463 | 0.427 | 0.534 | 0.281 | 0.196 | 0.447 | 0.342 | 0.274 | 0.474 |
| HAKE | 0.497 | 0.452 | 0.582 | 0.346 | 0.250 | 0.542 | 0.545 | 0.462 | 0.694 |
| SeAttE | **0.499** | **0.457** | **0.584** | **0.371** | **0.274** | **0.562** | **0.585** | **0.513** | **0.714** |

Table 3 shows the comparison between SeAttE and deep learning models. The table shows that SeAttE also achieves the best performance on WN18RR and YAGO3-10. Compared with the best deep learning model, SeAttE still has significant improvements: on YAGO3-10, MRR increases by 5.8%; on WN18RR, MRR increases by 3.2%. Nathani's model still keeps the best performance on FB15K-237, because it applies a novel attention-based feature embedding that captures both entity and relation features in any given entity's neighborhood. Utilizing graph neural network techniques for link prediction is our ongoing research.

**Table 3.** This is the comparison between SeAttE and deep learning models on WN18RR, FB15K-237 and YAGO3-10. The best results of each metric for each dataset are marked in bold.

| | WN18RR | | | FB15K-237 | | | YAGO3-10 | | |
|---|---|---|---|---|---|---|---|---|---|
| | MRR | Hit@1 | Hit@10 | MRR | Hit@1 | Hit@10 | MRR | Hit@1 | Hit@10 |
| ConvE | 0.427 | 0.390 | 0.508 | 0.305 | 0.219 | 0.476 | 0.488 | 0.399 | 0.658 |
| ConvKB | 0.249 | 0.056 | 0.525 | 0.230 | 0.140 | 0.415 | 0.420 | 0.322 | 0.605 |
| ConvR | 0.467 | 0.437 | 0.527 | 0.346 | 0.256 | 0.526 | 0.527 | 0.446 | 0.673 |
| CapsE | 0.415 | 0.337 | 0.560 | 0.160 | 0.073 | 0.356 | 0.000 | 0.00 | 0.00 |
| RSN | 0.280 | 0.198 | 0.444 | 0.395 | 0.346 | 0.483 | 0.511 | 0.427 | 0.664 |
| Nathani's | 0.440 | 0.361 | 0.581 | **0.518** | **0.460** | **0.626** | - | - | - |
| SeAttE | **0.499** | **0.457** | **0.584** | 0.371 | 0.278 | 0.562 | **0.585** | **0.513** | **0.714** |

Table 4 shows the comparison between SeAttE and tensor decomposition models. The table shows that SeAttE also achieves the best performance among all datasets. On WN18RR, RESCAL-DURA initially achieved the best performance. SeAttE achieves the same inference performance as the RESCAL-DURA model. On FB15K-237 and YAGO3-10, ComplEx-DURA initially performed the best inference. SeAttE achieves the same inference performance as ComplEx-DURA. This experiment also verifies the novelty of SeAttE and the proof in Section 4.2.

**Table 4.** This is the results of tensor decomposition models on WN18RR, FB15K-237 and YAGO3-10. The best results of each metric for each dataset are marked in bold.

| | WN18RR | | | FB15K-237 | | | YAGO3-10 | | |
|---|---|---|---|---|---|---|---|---|---|
| | MRR | Hit@1 | Hit@10 | MRR | Hit@1 | Hit@10 | MRR | Hit@1 | Hit@10 |
| CP | 0.438 | 0.414 | 0.485 | 0.333 | 0.247 | 0.508 | 0.567 | 0.494 | 0.698 |
| RESCAL | 0.455 | 0.419 | 0.493 | 0.353 | 0.264 | 0.528 | 0.566 | 0.490 | 0.701 |
| ComplEx | 0.460 | 0.428 | 0.522 | 0.346 | 0.256 | 0.525 | 0.573 | 0.500 | 0.703 |
| DisMult | 0.433 | 0.397 | 0.502 | 0.313 | 0.224 | 0.490 | 0.501 | 0.413 | 0.661 |
| SimplE | 0.398 | 0.383 | 0.427 | 0.179 | 0.100 | 0.344 | 0.453 | 0.358 | 0.632 |
| ANALOGY | 0.366 | 0.358 | 0.380 | 0.202 | 0.126 | 0.354 | 0.283 | 0.192 | 0.457 |
| HolE | 0.432 | 0.403 | 0.488 | 0.303 | 0.214 | 0.476 | 0.502 | 0.418 | 0.652 |
| TuckER | 0.459 | 0.430 | 0.510 | 0.352 | 0.259 | 0.536 | 0.544 | 0.466 | 0.681 |
| AutoSF | 0.490 | 0.451 | 0.567 | 0.360 | 0.267 | 0.552 | 0.571 | 0.501 | **0.715** |
| CP-DURA | 0.478 | 0.441 | 0.552 | 0.367 | 0.272 | 0.555 | 0.579 | 0.506 | 0.709 |
| RESCAL-DURA | 0.498 | 0.455 | 0.577 | 0.368 | 0.276 | 0.550 | 0.579 | 0.505 | 0.712 |
| ComplEx-DURA | 0.491 | 0.449 | 0.571 | 0.371 | 0.276 | 0.560 | 0.584 | 0.511 | 0.713 |
| SeAttE | **0.499** | **0.457** | **0.584** | **0.372** | **0.276** | **0.562** | **0.585** | **0.513** | 0.714 |

In summary, SeAttE belongs to the family of tensor decomposition models. Compared to other tensor models, SeAttE reaches the upper-performance limit of this family of models. SeAttE achieves the best performance as a tensor decomposition model compared with geometric models. SeAttE achieves the best performance on some datasets compared with deep learning models. Since Nathani's model utilizes a novel attention-based feature embedding that captures neighborhood features, it achieves the best performance in FB15K-237. Comparative experiments show that this operation of separating attribute space allows the model to focus on learning the semantic equivalence between relations, resulting in better performance approaching the theoretical limit.

### 5.3. Visualization and Analysis

In this part, we analyze the performance of SeAttE from three aspects. First, we visualize the embedding through T-SNE; then, we randomly select a pair of samples to analyze the function of SFBR and show the additional resources occupied by SFBR.

**Visualization.** We use T-SNE to visualize embeddings of tails. Suppose the link prediction task is $(h, r, ?)$, where $h$ and $r$ are head entities and relations, respectively. We randomly select ten queries in FB15k-237, each of which has more than 50 answers. Then we use T-SNE to visualize the embeddings generated by RESCAL and SeAttE. For each question, we convert the answers into two-dimensional points with T-SNE and display them on the graph with the same color.

As shown in Figures 5 and 6, it is a visualization of the distribution of answers to 10 questions. SeAttE makes the answers to the same question more similar, indicating that SeAttE effectively separates the needed semantics of each entity and suppresses the attributes of other dimensions, which verifies the claim in Section 4.1.

**Resource occupation.** As shown in Tables 5–7, we compare the parameter size of different models under the identical dimension of entities. When the entity vector dimension $d$ is fixed, the number of parameters in SeAttE increases slightly as the dimension $k$ of each subspace increases. First, we compare the parameters of ComplEx and SeAttE. When

the subspace dimension $k$ is set to two, the parameters of SeAttE and ComplEx are the same, which is consistent with the proof in Section 4.2. We find that the parameter amount of SeAttE is slightly higher than that of ComplEx as the subspace dimension $k$ increases. Then we compare the parameters of RESCAL and SeAttE in the three tables. We find that the parameter amount of SeAttE is much lower than that of RESCAL at the same entity dimension. In summary, the experiments show that learning too many parameters for the attribute space separation task in traditional tensor decomposition models is transformed into the structure's design in SeAttE. SeAttE achieves good performance while significantly reducing the number of parameters, verifying the statement in Section 4.1.

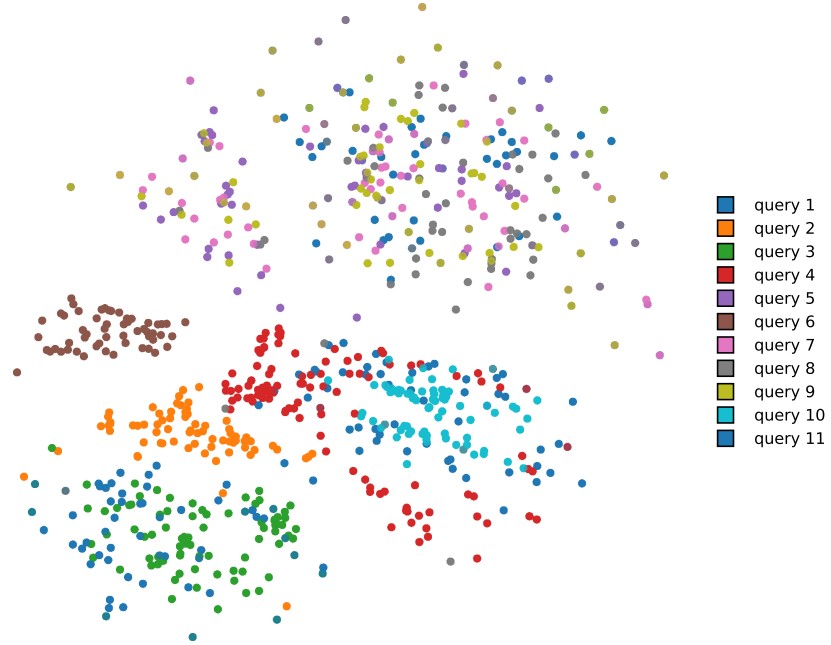

**Figure 5.** Visualization of tail entities in RESCAL using T-SNE. A point represents a tail entity. Points in the same color represent tail entities that have the same context $\left( h_r, r_j \right)$.

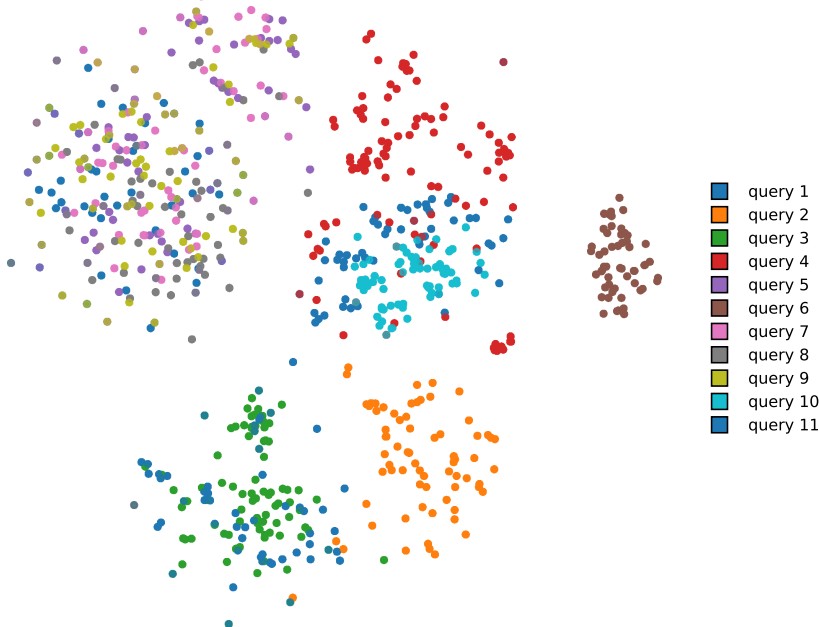

**Figure 6.** Visualization of tail entities in SeAttE using T-SNE.

**Table 5.** This is the comparison of parameters between ComplEx, RESCAL and SeAttE, when entities have the same dimension ($d = 1500$). $k$ denotes the dimension of each attribute subspace in SeAttE.

| Model | WN18RR | FB15K-237 | YAGO3-10 |
|---|---|---|---|
| ComplEx | 61.48 M | 23.23 M | 185.00 M |
| RESCAL | 110.98 M | 1089.73 M | 351.50 M |
| SeAttE ($k = 1$) | 61.45 M | 22.52 M | 184.89 M |
| SeAttE ($k = 2$) | 61.48 M | 23.23 M | 185.00 M |
| SeAttE ($k = 4$) | 61.54 M | 24.65 M | 185.23 M |
| SeAttE ($k = 8$) | 61.61 M | 27.43 M | 185.37 M |

**Table 6.** This is the comparison of parameters between ComplEx, RESCAL and SeAttE, when entities have the same dimension ($d = 1000$).

| Model | WN18RR | FB15K-237 | YAGO3-10 |
|---|---|---|---|
| ComplEx | 40.99 M | 15.49 M | 123.33 M |
| RESCAL | 62.99 M | 489.49 M | 197.33 M |
| SeAttE ($k = 1$) | 40.97 M | 15.02 M | 123.26 M |
| SeAttE ($k = 2$) | 40.99 M | 15.49 M | 123.34 M |
| SeAttE ($k = 4$) | 41.03 M | 16.44 M | 123.48 M |
| SeAttE ($k = 8$) | 41.11 M | 18.33 M | 123.78 M |

**Table 7.** This is the comparison of parameters between ComplEx, RESCAL and SeAttE, when entities have the same dimension ($d = 500$).

| Model | WN18RR | FB15K-237 | YAGO3-10 |
|---|---|---|---|
| ComplEx | 20.49 M | 7.74 M | 61.67 M |
| RESCAL | 25.99 M | 126.24 M | 80.17 M |
| SeAttE ($k = 1$) | 20.48 M | 7.50 M | 61.63 M |
| SeAttE ($k = 2$) | 20.49 M | 7.74 M | 61.66 M |
| SeAttE ($k = 4$) | 20.51 M | 8.22 M | 61.74 M |
| SeAttE ($k = 8$) | 20.59 M | 9.09 M | 61.88 M |

## 6. Conclusions and Future Work

We investigate the design approaching the theoretical performance of tensor decomposition models in this paper. SeAttE is based on the observation that judging the rationality of a particular triple is to compare specific attributes between the entities, ignoring other unrelated dimensions. The comparison of triples should first separate the properties that need to be compared. Therefore, we provide SeAttE—a tensor decomposition model based on separating attribute space for knowledge graph completion in this paper. SeAttE is the first model among the tensor decomposition family to consider the attribute space separation task. Furthermore, SeAttE transforms the learning of too many parameters for the attribute space separation task to the structure's design. This operation allows the model to focus on learning the semantic equivalence between relations, causing the performance to approach the theoretical limit. Experiments show that SeAttE can achieve the best performance among the traditional tensor decomposition models. The visualization shows that SeAttE can effectively extract the relevant dimensions and distinguish the comparisons among different attributes. Compared with the RESCAL, the resource occupation of SeAttE is much lower than that of RESCAL. Compared with the ComplEx, SeAttE only has a slight growth in resource occupation.

Recently, graph neural networks have achieved good performance on link prediction. In the future, we plan to evaluate SeAttE on more datasets and leverage the graph attention framework to capture higher-order relations between entities.

**Author Contributions:** Conceptualization, Z.L. and J.Y.; validation, K.H.; formal analysis, Z.L.; investigation, H.L.; resources, L.C.; writing—original draft preparation, L.Q. and X.L. All authors have read and agreed to the published version of the manuscript.

**Funding:** This research was funded by Anhui Provincial Natural Science Foundation OF FUNDER grant number No. 1908085MF202 and and Independent Scientific Research Program of National University of Defense Science and Technology OF FUNDER grant number No. ZK18-03-14.

**Informed Consent Statement:** Not applicable.

**Data Availability Statement:** MDPI Research Data Policies at https://github.com/ibalazevic/TuckER.

**Acknowledgments:** This work was partially supported by the Anhui Provincial Natural Science Foundation (No. 1908085MF202) and Independent Scientific Research Program of National University of Defense Science and Technology (No. ZK18-03-14).

**Conflicts of Interest:** The authors declare no conflict of interest.

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
