# Peer review of "SeAttE: An Embedding Model Based on Separating Attribute Space for Knowledge Graph Completion"

_electronics, doi:10.3390/electronics11071058_

Round 1

Reviewer 1 Report

The paper explores the plan moving toward the hypothetical exhibition of tensor disintegration models. The paper is well written and presents a novel model. Following changes are required

  1. Explain the role of separating attribute space.
  2. The paper believes that we should first separate the properties that need to be compared. How the paper will believe. The authors can believe that the properties need to be compared. Therefore, the introduction section requires a massive re-write-up.
  3. This paper proves that RESCAL, DisMult, and ComplEx are all special cases of SeAttE. The paper will not prove, its authors will prove.
  4. Add the structure of the paper in the introduction section. For example, the rest of the paper is structured as follows:
  5. Related work is very poor. It requires more up-to-date references at least 15 more references for tensor decomposition models, geometric models. and deep learning models. Once the papers are added. Add the comparison paragraph that how your approach is different than the proposed works.
  6. Section 4.1 need to be explained before you start 4.1.1 or just exclude 4.1.1.
  7. Did you compare parameters between ComplEx and SeAttE with different dimensions
  8. Section 5.2 needs to be revised as the main results are quite a vague name
  9. Finally, add future work to the conclusions.

Reviewer 2 Report

There are many link prediction algorithms exist autors should show comparision with most of them.

What is novelty in paper?

What is the application is not properly mentioned?

Validate methods are not mentioned.

I feel lot of work is already done in this area .. Authors have not done proper literature survey.

English related flaws are many.

Pipeline diagram is not clear.

Data sets can be many which can be used further.

Reviewer 3 Report

The general idea of the paper seems to be good. However, the paper organization is not acceptable, and there are several major technical challenges that should be effectively addressed.
Comments:
1- The abstract has been briefly written and should be enriched by adding the main ideas and contributions.
2- There are some grammatical errors and typos that should be corrected before publication.
3- It is recommended to provide a nomenclature at the beginning of the paper to define all variables clearly.
4- The introduction has been vaguely written. This section should highlight the problem statement, contribution and motivation.  

5. The outlines of the paper must be written as the last paragraph of the introduction section

6- The main contribution of the paper should be highlighted and emphasized. It would be great if the drawbacks and gaps of literature are clear and, particularly, how the proposed approach aims at filling these gaps.

7. Simulation Results should be enriched. More detailed scenarios should be added. This section is very weak.
8- A separate section should be added for discussion of obtained results and main achievements.
9- The quality of all the figures is poor and should be improved.
10- References should be prepared according to the journal referencing style

Round 2

Reviewer 1 Report

The authors did not provide the highlighted version. Also, they did not list the changes in the cover letter using line numbers and paragraph numbers. I can not track the changes

Reviewer 2 Report

How to know which changes are done in the manuscript.

Reviewer 3 Report

The authors revised the paper very well and have addressed all comments.

I recommend accepting the paper at this round.

Author Response

Thanks very much for your comments and help.

Round 3

Reviewer 1 Report

The authors have addressed most of the comments. But I am still not satisfied with related work. 

It requires more references in Model-Based on Separating Attribute
Space. I suggest adding at least 10 up-to-date references and also increasing the related work section to 1 and a half pages.

Reviewer 2 Report

i am not convinced with the novelty part.

Author Response

This manuscript is a resubmission of an earlier submission. The following is a list of the peer review reports and author responses from that submission.